# THINK TWICE, ACT ONCE: TOKEN-AWARE COMPRESSION AND ACTION REUSE FOR EFFICIENT INFERENCE IN VISION-LANGUAGE-ACTION MODELS

## ABSTRACT

Vision-Language-Action (VLA) models have emerged as a powerful paradigm for robot control through natural language instructions. However, their high inference cost—stemming from large-scale token computation and autoregressive decoding—poses significant challenges for real-time deployment and edge applications. While prior work has primarily focused on efficient architectural optimization, we take a different and innovative perspective by identifying a dual form of redundancy in VLA models: (i) high similarity across consecutive action steps, and (ii) substantial redundancy in visual tokens. Motivated by these observations, we propose FLASHVLA, the first training-free and plug-and-play acceleration framework that enables action reuse in VLA models. Specifically, FLASHVLA improves inference efficiency through a token-aware action reuse mechanism that avoids redundant decoding across stable action steps, and an information-guided visual token selection strategy that prunes low-contribution tokens. Extensive experiments on the LIBERO benchmark show that FLASHVLA reduces FLOPs by 55.7% and latency by 36.0%, with only a 0.7% drop in task success rate. These results demonstrate the effectiveness of FLASHVLA in enabling lightweight, low-latency VLA inference without retraining.

## 1 INTRODUCTION

In the development of embodied intelligence systems, Vision-Language-Action (VLA) models are rapidly emerging as a key technology for enabling general-purpose behavior control. By integrating visual perception, language understanding, and action generation, VLA models empower embodied agents to execute complex tasks based on natural language instructions, demonstrating strong generalization and task adaptability Brohan et al. (2022); Nair et al. (2022); Bai et al. (2023); Chen et al. (2023); Cheang et al. (2024); Li et al. (2023); Jiang et al. (2023); Kim et al. (2024); Chen et al. (2024d); Chi et al. (2023); Duan et al. (2024); Singh et al. (2023). However, despite their impressive performance in task execution, VLA models often involve heavy computational loads and high latency during inference, making them difficult to deploy in high-frequency control settings and limiting their applicability in more dexterous and complex bimanual manipulation tasks Kim et al. (2025); Liu et al. (2024b); Wen et al. (2025); Li et al. (2024a).

Current VLA architectures typically fall into two paradigms: the autoregressive generation paradigm, such as OpenVLA Kim et al. (2024), which encodes multimodal inputs into tokens and decodes actions step-by-step using a language model; and the diffusion policy paradigm Wen et al. (2024); Li et al. (2024b); Yan et al. (2024); Chi et al. (2023); Hou et al. (2024), which formulates action generation as a conditional denoising process and enables parallel trajectory sampling. Both paradigms rely heavily on Transformer architectures, where each inference step incurs a quadratic complexity $\mathcal{O}(N^2)$ with respect to sequence length $N$, leading to high computational cost. To mitigate this, recent work focuses on architectural-level

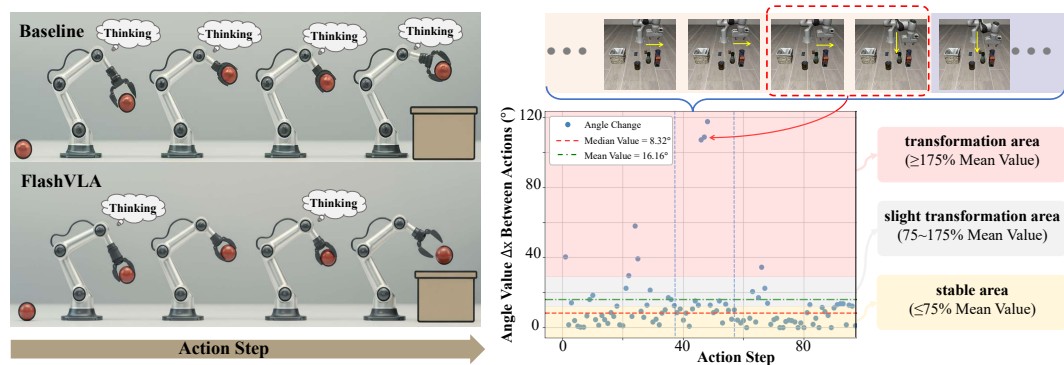

Figure 1: Motivation behind our proposed FLASHVLA. (Left) Illustration of action computation frequency: the baseline triggers frequent "thinking" events, while FLASHVLA reduces such computations. (Right) Analysis of action-transition dynamics: the figure shows the change in the VLA model's output vector at each time step relative to the previous one; the vertical axis indicates the directional difference between consecutive actions, with most steps in the stable area and only a few entering transformation regions.

optimizations, including action chunking Song et al. (2025); Liu et al. (2024d), parallel decoding Kim et al. (2025), low-rank adaptation Wen et al. (2025), and model quantization Park et al. (2024). While effective, these methods typically require additional training overhead.

Unlike prior work that focuses on architectural optimizations, we take a new perspective by analyzing the temporal behavior of VLA models at the action level. We observe that in many tasks (e.g., OpenVLA Kim et al. (2024)), **consecutive action steps show minimal directional change, suggesting semantic redundancy** (see Fig. 1). This indicates that actions in stable phases can be reused to avoid redundant computation. We further find **significant redundancy in visual tokens** (see Appendix F), consistent with observations in vision-language models (VLMs) Chen et al. (2024c); Yang et al. (2024); Zhang et al. (2025); Chen et al. (2024a), which reveals potential for reducing computational cost along another computational dimension.

Building on these observations, we propose FLASHVLA, a training-free, plug-and-play dual-path acceleration framework that improves VLA inference efficiency via action reuse and token pruning. The first component is a **token-aware reuse mechanism** that compares both action similarity and visual token stability to decide whether to skip computation and reuse the previous action. The second is a **visual token selection strategy** based on information contribution scores, retaining informative tokens while discarding low-impact ones. The proposed FLASHVLA integrates seamlessly with Flash Attention-based VLA models and follows a "**Think Twice, Act Once**" paradigm: it performs lightweight assessment before execution, selecting between skipped execution and lightweight execution, as illustrated in Fig. 1 by reducing the frequency of action computations, significantly reducing FLOPs and latency while maintaining control accuracy.

To evaluate the effectiveness of FLASHVLA, we conduct systematic experiments on the LIBERO benchmark using OpenVLA as the primary backbone. To further examine generality across different architectures and environments, we additionally validate on UniVLA and VLAbench. FLASHVLA achieves a 55.7% reduction in FLOPs and a 36.0% reduction in latency without any additional training, while reducing the number of visual tokens to 62.5% of the original input. Notably, the average success rate drops by only 0.7% compared to the baseline VLA model. Ablation studies and benchmark results collectively demonstrate that FLASHVLA significantly reduces computational cost while preserving task performance, enabling efficient VLA inference with minimal performance drop. Our key contributions are summarized as follows:

1. We identify a novel form of action-level and token-level redundancy in VLA inference. Specifically, we observe that most consecutive action steps yield highly similar outputs with only minor directional

changes, allowing action reuse in stable phases. Additionally, many visual tokens contribute little to the inference process, revealing a degree of visual redundancy similar to that observed in MLLMs.

2. We introduce FLASHVLA, the first training-free and plug-and-play acceleration framework that enables action reuse in VLA models. It integrates a token-aware action reuse mechanism to skip redundant computation in stable action steps, and a visual token selection strategy based on information contribution scores to retain informative tokens. It is worth noted that FLASHVLA is fully compatible with Flash Attention-based VLA backbones.

3. We conduct comprehensive experiments on four representative tasks from the LIBERO benchmark. When visual tokens are reduced to 62.5% of the original input, FLASHVLA lowers inference latency by 36.0% and decreases the FLOPs of visual token computation by 55.7%, while incurring only a 0.7% drop in success rate. These results demonstrate that FLASHVLA achieves significant efficiency gains with limited performance trade-off.

## 2 RELATED WORK

Recent advances in VLA models highlight the critical role of architecture in determining both performance and efficiency. To address the computational challenges inherent in these models, a variety of acceleration methods Kim et al. (2025); Song et al. (2025); Li et al. (2024b); Wen et al. (2025); Liu et al. (2024a); Xu et al. (2025); Yue et al. (2024); Park et al. (2024) have been proposed across two dominant paradigms: autoregressive generation and diffusion policy Zitkovich et al. (2023); Kim et al. (2024); Black et al.; Liu et al. (2024b); Wen et al. (2024); Yan et al. (2024); Li et al. (2024b); Chen et al. (2023).

**Vision-language-action models.** Vision-language-action (VLA) models provide a promising direction for training generalist robot policies Ahn et al. (2022); Brohan et al. (2022); Black et al.; Duan et al. (2024) and are built on large-scale robot learning datasets Liu et al. (2023); O'Neill et al. (2024); Fang et al. (2023); Khazatsky et al. (2024); Li et al. (2024c). Most VLA models follow one of two paradigms: autoregressive generation and diffusion policy. Autoregressive models, such as RT-2 Zitkovich et al. (2023) and OpenVLA Kim et al. (2024), encode multimodal inputs into tokens and decode actions step-by-step using a language model. RT-2 treats actions as text tokens and trains them alongside natural language, while OpenVLA combines a vision backbone with a language model trained on large-scale robot trajectories. Pi0 Black et al. is another autoregressive model that uses flow matching for faster action generation. Diffusion-based models, including RDT-1B Liu et al. (2024b), Diffusion-VLA Wen et al. (2024), DNACT Yan et al. (2024), and CogACT Li et al. (2024b), formulate action generation as conditional denoising. Diffusion-VLA combines autoregressive and diffusion methods to improve robustness. DNACT focuses on multi-task 3D policy learning, while CogACT generates diverse action sequences to improve flexibility. However, the large size of VLA models leads to high computational cost, limiting real-time deployment and high-frequency control.

**Acceleration for VLA models.** Existing methods mainly focus on architectural-level optimizations tailored to the two main VLA paradigms, including action chunking Song et al. (2025); Kim et al. (2025), which splits complex actions into smaller segments to reduce per-step computation; parallel decoding Song et al. (2025); Kim et al. (2025), which enables simultaneous generation of multiple actions; low-rank adaptation Wen et al. (2025); Hu et al. (2022), which compresses model weights to reduce parameters; and model quantization Pertsch et al. (2025); Park et al. (2024), which lowers numerical precision to save memory and computation. While these techniques improve efficiency, they require additional training or fine-tuning. Training-free acceleration remains underexplored. Although some pruning methods from VLMs Chen et al. (2024b); Zhang et al. (2024b); Yang et al. (2024); Liu et al. (2024c) are training-free, they are incompatible with Flash Attention and do not sufficiently consider the structural characteristics of VLA models. To address these limitations, we propose a training-free, plug-and-play dual-path framework that accelerates VLA inference through action reuse and token pruning.

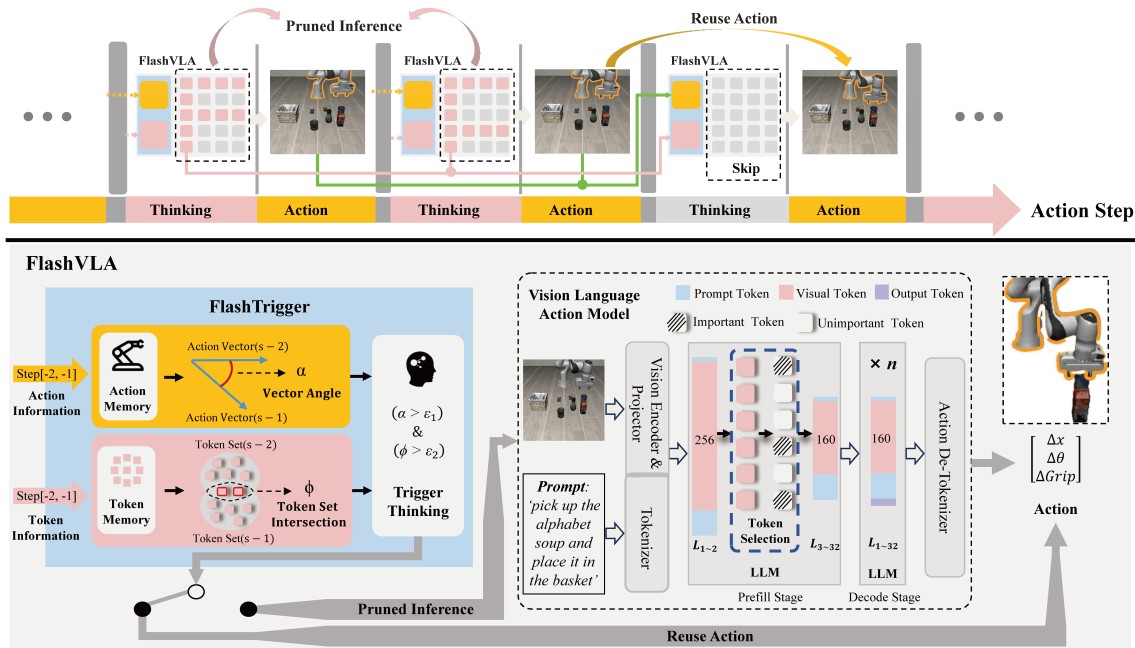

Figure 2: Framework of our FLASHVLA. We illustrate how our method works as action steps change. Before each inference, FlashTrigger will think about whether it can reuse the output of the previous action based on action memory and token memory (as shown in blue block). If the trigger condition is met, this inference is skipped. If the trigger condition is not met, proceed to the pruned inference step. In pruned inference step, we select the set of important visual tokens in the prefill stage and prune the other unimportant tokens. After inference, action information and token information are used to update action memory and token memory.

## 3 FLASHVLA

The FLASHVLA architecture is illustrated in Fig.2. In the following, we first introduce the standard formulation of VLA models in Section3.1. We then detail our visual token selection strategy and action reuse mechanism in Section 3.2 and Section 3.3, respectively. The overall algorithmic flow of FLASHVLA is provided in Appendix A (Algorithm 1).

### 3.1 PRELIMINARIES

VLA models extend vision-and-language foundation models for robotic control by generating actions from visual inputs and language prompts. A representative example is OpenVLA Kim et al. (2024), a 7B-parameter open-source model that establishes a strong baseline for general-purpose manipulation. It consists of a vision encoder that combines DINOv2 Oquab et al. (2023) and SigLIP Zhai et al. (2023) features, a projector that maps visual features into the language embedding space, and a LLaMA Touvron et al. (2023) language model as the backbone. Given an image and a text prompt, the encoder and projector produce a sequence of visual tokens $T^v = \{T_1^v, T_2^v, ...T_N^v\}$, while the prompt is tokenized into $T^l = \{T_1^l, T_2^l, ..., T_M^l\}$. These tokens are concatenated and passed to the language model, which autoregressively generates actions as control outputs. However, the large number of visual tokens, combined with highly repetitive action outputs, leads to significant computational overhead. These observations highlight two key sources of inefficiency in VLA

models: visual token redundancy and temporal redundancy in action generation. In the following sections, we introduce methods to address both aspects and improve inference efficiency without additional training.

### 3.2 VISUAL TOKEN SELECTION STRATEGY VIA INFORMATION CONTRIBUTION THEORY

We observe that VLA models exhibit visual token redundancy patterns similar to those found in VLMs Chen et al. (2024c), where attention distributions are highly sparse beyond the initial few transformer layers (see Appendix F). This motivates our token selection strategy based on information contribution theory, which identifies tokens that best preserve the structure of the visual feature space. However, most recent architectures (e.g., OpenVLA) rely on Flash Attention Dao et al. (2022), making traditional attention score-based selection infeasible. To address this, we directly operate on the attention output matrix and rank tokens by their estimated contribution to the global feature representation.

Let $\hat{T}^v \in \mathbb{R}^{N \times d}$ denote the attention output matrix corresponding to the visual tokens, where $N$ is the number of tokens and $d$ is the hidden dimension. To quantify the amount of information contained in $\hat{T}^v$, we perform singular value decomposition (SVD), where $\hat{T}^v = U\Sigma V^\top$. Here, $U \in \mathbb{R}^{N \times N}$ contains the left singular vectors, $V \in \mathbb{R}^{d \times d}$ contains the right singular vectors, and $\Sigma \in \mathbb{R}^{N \times d}$ is a diagonal matrix of singular values $\sigma_i$ sorted in descending order. Each token representation $\hat{T}^v(x)$ corresponds to a row of $\hat{T}^v$ and can be expressed as:

$$\hat{T}^v(x) = \sum_{i=1}^{r} u_{xi}\sigma_i v_i^\top \tag{1}$$

where $r$ is the effective rank of the matrix. We define the *information contribution score* (ICS) of the $x$-th token as:

$$C(x) = \sum_{i=1}^{r} |u_{xi}\sigma_i| \tag{2}$$

This score measures the magnitude of the token's projection onto the dominant singular directions, weighted by their corresponding singular values. Tokens with higher $C(x)$ values are expected to contribute more significantly to the overall representation. As shown in Fig. 3, ICS-selected tokens can better focus on information-rich regions compared with random selection. A theoretical justification of ICS is provided in Appendix C.

### 3.3 TOKEN-AWARE ACTION REUSE STRATEGY

Action outputs in VLA models often exhibit minimal variation or remain identical across frames (Fig.1), and can be regarded as redundant actions suitable for direct reuse to accelerate inference. As shown in Fig.2, FlashTrigger decides whether to reuse the previous action or perform a new pruned inference. It consists of Action Memory, Token Memory, and a Trigger Thinking module. To ensure stability, reuse is not applied in the first two frames; the current frame is denoted as the $s$-th with $s > 2$. At the action level, consistency is measured by the variation in action vectors $\vec{A}$. Action Memory stores the outputs from the previous two frames, $\vec{A}(s-2)$ and $\vec{A}(s-1)$, and computes the angle $\alpha$ between them to quantify the change:

$$\alpha(s) = Arccos(\frac{\vec{A}(s-2) \cdot \vec{A}(s-1)}{||\vec{A}(s-2)|| \times ||\vec{A}(s-1)||}) \tag{3}$$

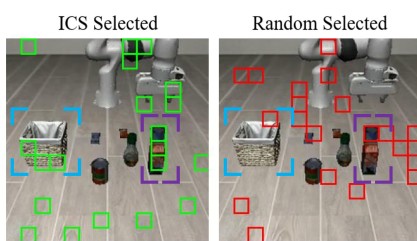

ICS Selected      Random Selected

*Prompt: pick up the aphabet soup and place it in the basket.*

Figure 3: Comparison of token selection strategies on a robot-view image. Left: selection by the proposed ICS method. Right: uniform random selection. ICS typically focuses on semantically meaningful and information-dense regions.

In token level, the change of set $I$ computed in Section 3.2 is used to determine the degree of change in environment with a VLA model perspective. The Token Memory dynamically updates and stores the computed $I$ from the previous two frames: $I(s-2)$ and $I(s-1)$. We calculate the intersection ratio $\phi$ between $I(s-2)$ and $I(s-1)$ to determine the extent of the change of $I$:

$$\phi(s) = \frac{Len\left[I(s-2) \cap I(s-1)\right]}{Len\left[I(s-1)\right]} \tag{4}$$

where $Len(\cdot)$ returns the number of elements in the set.

At the same time, the lower limit of $\alpha(s)$ change is denoted by $\varepsilon_1$. We define $\delta$ as the maximum number of allowed changes in the set of visual tokens at the two action steps before and after. Thus the lower limit of the threshold $\varepsilon_2$ at which $\phi(s)$ is allowed to change can be defined:

$$\varepsilon_2 = 1 - \frac{\delta}{Len\left[I(s-1)\right]} \tag{5}$$

Before each inference, the Trigger Thinking module thinks ahead about whether this inference should directly reuse $\vec{A}(s-1)$ or recalculate how to act. It can be represented as:

$$Trigger\ Thinking = \begin{cases} Reuse\ Action, & if\ \alpha(s) > \varepsilon_1\ and\ \phi(s) > \varepsilon_2 \\ Pruned\ Inference, & else \end{cases} \tag{6}$$

When the reuse condition is not met, VLA performs pruned inference using the informative visual token set $I$ instead of the full token set. In other words, FLASHVLA consistently accelerates the inference process, regardless of whether the action is reused.

Table 1: Main results of FLASHVLA under different visual token budgets across four task suites in the LIBERO. We report SR, visual-token FLOPs, and latency at five token settings. With 160 visual tokens, FLASHVLA achieves a strong balance between SR and efficiency—reducing FLOPs by 55.7% and latency by 36.0%—while maintaining comparable or even improved success rates across most tasks.

| Task / Visual token | | 256 (Baseline) | 192 (75%) | 160 (62.5%) | 128 (50%) | 96 (37.5%) |
|---|---|---|---|---|---|---|
| **LIBERO-Spatial** | SR (%) | 84.2 | 81.8 ($-$ 2.4) | 82.6 ($-$ 1.6) | 75.4 ($-$ 8.8) | 67 ($-$ 17.2) |
| | Flops ($10^{12}$) | 1.31 | 0.8 ($\downarrow$ 38.9%) | 0.66 ($\downarrow$ 49.6%) | 0.51 ($\downarrow$ 61.1%) | 0.43 ($\downarrow$ 67.2%) |
| | Latency (ms) | 82.7 | 62.7 ($\downarrow$ 24.2%) | 61.2 ($\downarrow$ 26.0%) | 58.1 ($\downarrow$ 29.7%) | 58.9 ($\downarrow$ 28.8%) |
| **LIBERO-Object** | SR (%) | 86.4 | 86.6 ($+$ 0.2) | 86.6 ($+$ 0.2) | 85.2 ($-$ 1.2) | 83.6 ($-$ 2.8) |
| | Flops ($10^{12}$) | 1.31 | 0.74 ($\downarrow$ 43.5%) | 0.57 ($\downarrow$ 56.5%) | 0.42 ($\downarrow$ 67.9%) | 0.33 ($\downarrow$ 74.8%) |
| | Latency (ms) | 82.7 | 58.8 ($\downarrow$ 28.9%) | 53.1 ($\downarrow$ 35.8%) | 45.3 ($\downarrow$ 45.2%) | 47.2 ($\downarrow$ 42.9%) |
| **LIBERO-Goal** | SR (%) | 75.4 | 76.2 ($+$ 0.8) | 78.8 ($+$ 3.4) | 76.8 ($+$ 1.4) | 70.2 ($-$ 5.2) |
| | Flops ($10^{12}$) | 1.31 | 0.71 ($\downarrow$ 45.8%) | 0.6 ($\downarrow$ 54.2%) | 0.49 ($\downarrow$ 62.6%) | 0.37 ($\downarrow$ 71.8%) |
| | Latency (ms) | 82.7 | 55.2 ($\downarrow$ 33.3%) | 56.8 ($\downarrow$ 31.3%) | 54.1 ($\downarrow$ 34.6%) | 53.8 ($\downarrow$ 34.9%) |
| **LIBERO-Long** | SR (%) | 51.4 | 50.2 ($-$ 1.2) | 46.8 ($-$ 4.6) | 46.4 ($-$ 5.0) | 45.2 ($-$ 6.2) |
| | Flops ($10^{12}$) | 1.31 | 0.54 ($\downarrow$ 58.8%) | 0.47 ($\downarrow$ 64.1%) | 0.4 ($\downarrow$ 69.5%) | 0.32 ($\downarrow$ 75.6%) |
| | Latency (ms) | 82.7 | 42.43 ($\downarrow$ 48.7%) | 40.35 ($\downarrow$ 51.2%) | 38.31 ($\downarrow$ 53.7%) | 44.05 ($\downarrow$ 46.7%) |
| **Average** | SR (%) | 74.4 | 73.7 ($-$ 0.7) | 73.7 ($-$ 0.7) | 71.0 ($-$ 3.4) | 66.5 ($-$ 7.9) |
| | Flops ($10^{12}$) | 1.31 | 0.70 ($\downarrow$ 46.6%) | 0.58 ($\downarrow$ 55.7%) | 0.46 ($\downarrow$ 64.9%) | 0.36 ($\downarrow$ 72.5%) |
| | Latency (ms) | 82.7 | 54.8 ($\downarrow$ 33.7%) | 52.9 ($\downarrow$ 36.0%) | 49.0 ($\downarrow$ 40.7%) | 51.0 ($\downarrow$ 38.3%) |

## 4 EXPERIMENT

### 4.1 EXPERIMENTAL SETUP

We evaluate FLASHVLA on the LIBERO benchmark using OpenVLA, following standard protocols. Detailed experimental environment, implementation settings, and hyperparameters are provided in Appendix D.

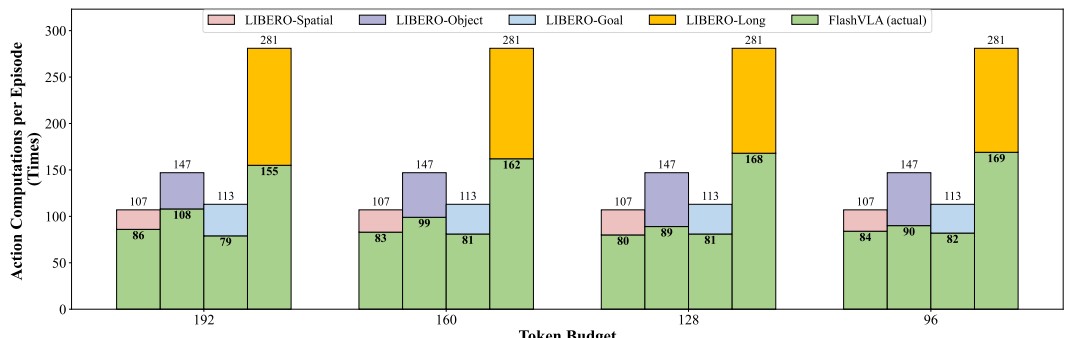

Figure 4: Average number of action computations per episode under different token budgets on LIBERO tasks. FlashVLA consistently requires fewer actions than the 256-token baseline.

## 4.2 MAIN RESULTS ON LIBERO BENCHMARK

We evaluate the performance of FLASHVLA across four task suites in the LIBERO benchmark under varying visual token budgets. As shown in Table 1, the 160-token configuration (62.5% of the original) offers the best accuracy–efficiency trade-off. Compared to the full 256-token baseline, it reduces visual-token FLOPs by 55.7% (from 1.31 to 0.58 $\times 10^{12}$) and lowers inference latency by 36.0% (from 82.7ms to 52.9ms), while maintaining the same average success rate (73.7% vs. 74.4%). Interestingly, we observe a slight gain at 160 tokens in both **LIBERO-Object** and **LIBERO-Goal**, where the success rate increases from 86.4% to 86.6% and from 75.4% to 78.8%, respectively. This suggests that modest token pruning may even help mitigate redundancy and stabilize policy execution. In contrast, **LIBERO-Long** is more sensitive to pruning, showing a larger SR decline when the token count drops below 160. Nevertheless, even in such cases, FLASHVLA achieves substantial savings in computational cost (e.g., 64.1% FLOPs reduction at 160 tokens).

These results highlight that FLASHVLA, particularly at the 160-token configuration, significantly improves inference efficiency without compromising performance, making it suitable for real-world deployment across diverse embodied tasks. Moreover, as illustrated in Fig. 4, FlashVLA consistently requires fewer action computations per episode compared to the 256-token baseline, demonstrating its holistic efficiency advantage. Further, we visualize the trajectories of FlashVLA and the baseline during successful task executions (Fig. 5), showing that FlashVLA exhibits smoother and more stable motion while achieving comparable outcomes.

Table 2: Ablation on the two core modules—Pruned Inference and Action Reuse—and the *ActionVector/TokenSet* components within the reuse mechanism. At the baseline of 256 tokens, SR is 84.2% and FLOPs are $1.31 \times 10^{12}$. FlashVLA achieves the best trade-off, while removing modules or components leads to either higher FLOPs or lower SR.

| Method | 192 Tokens | | 160 Tokens | | 128 Tokens | | 96 Tokens | |
|---|---|---|---|---|---|---|---|---|
| | SR (%) | FLOPs $(10^{12})$ | SR (%) | FLOPs $(10^{12})$ | SR (%) | FLOPs $(10^{12})$ | SR (%) | FLOPs $(10^{12})$ |
| **w/o Action Reuse** | 84.6 | 1.00 | 83.6 | 0.85 | 78.2 | 0.69 | 67.8 | 0.54 |
| **w/o Pruned Inference** | 82.2 | 1.04 | 80.6 | 1.01 | 81.0 | 0.97 | 80.2 | 0.97 |
| **w/o ActionVector** | 81.6 | 0.68 | 79.6 | 0.56 | 73.2 | 0.44 | 65.4 | 0.35 |
| **w/o TokenSet** | 76.4 | 0.52 | 74.8 | 0.44 | 73.4 | 0.37 | 62.2 | 0.29 |
| **FlashVLA** | 81.8 | 0.80 | 82.6 | 0.66 | 75.4 | 0.51 | 67.0 | 0.43 |

Table 3: Overall Evaluation Performance of FLASHVLA in VLAbench Zhang et al. (2024a) under different visual token budgets.

| Visual token | SR (%) | FLOPs $(10^{12})$ |
|---|---|---|
| 256 (baseline) | 7.0 | 1.31 |
| 192 | 8.0 | 0.62 |
| 160 | 5.0 | 0.62 |
| 128 | 6.0 | 0.41 |
| 96 | 7.0 | 0.30 |

### 4.3 ABLATION STUDY

**Effect of Token Pruning and Action Reuse.**   We evaluate the contribution of each FLASHVLA component via ablations on LIBERO-Spatial, disabling *Pruned Inference* or *Action Reuse*. As shown in Table 2, both are crucial for efficient inference with minimal performance loss. Removing Action Reuse preserves token pruning benefits and still reduces FLOPs (e.g., 0.66 at 160 tokens vs. 0.85), while maintaining similar SR, indicating that Action Reuse mainly improves efficiency. In contrast, disabling Pruned Inference yields consistently higher FLOPs with only marginal SR gains—for example, at 128 tokens, FLOPs rise from 0.51 (FlashVLA) to 0.97 ($\times 10^{12}$), while SR improves from 75.4% to 81.0%. The full FlashVLA combines both strategies and achieves the best performance–efficiency trade-off.

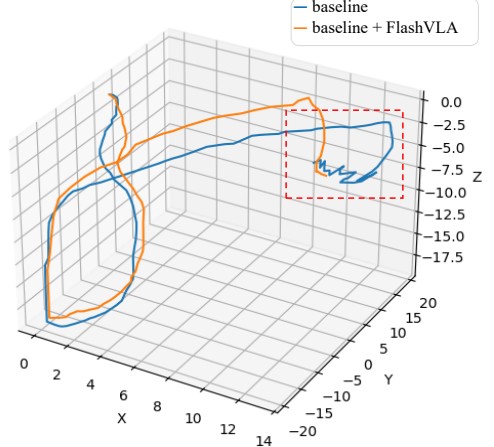
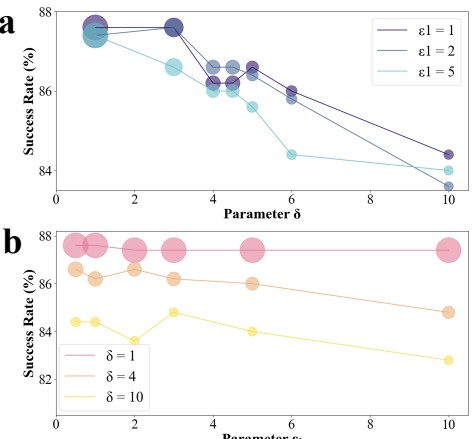

Figure 5: Trajectory of action. We visualize the trajectory of action in 3-dimensional space. The location of red dashed box illustrates the smoother trajectory of FLASHVLA for the same task.

Figure 6: Hyperparameter Sensitivity. Experiments are conducted to investigate the effect of parameter $\varepsilon_1$ and $\delta$. The size of point represents the size of the FLOPs.

**Component Analysis of Action Reuse Module.**   We assess the Action Reuse module in FLASHVLA via ablations on LIBERO-Spatial, removing either the action vector (*w/o ActionVector*) or token information (*w/o TokenSet*). As shown in Table 2, disabling either degrades performance. Without the action vector, SR drops under tight budgets (e.g., 82.6% $\rightarrow$ 65.4% at 96 tokens), showing temporal consistency is critical. Without token stability, the model reuses too aggressively, yielding lowest FLOPs (0.29 at 96 tokens) but unstable control (SR 62.2%). These results underscore their complementarity: the action vector ensures temporal continuity, while token stability reflects environmental change, jointly balancing efficiency and robustness.

**Hyperparameter Sensitivity.**   We conduct experiments to analyze the impact of two hyperparameters, $\varepsilon_1$ and $\delta$, on performance and efficiency, as shown in Fig.6. Fig.6(a) shows that varying $\varepsilon_1$ has negligible effect on both success rate and FLOPs, indicating that our method is largely insensitive to this parameter. In contrast, Fig. 6(b) shows that $\delta$ significantly affects both metrics: increasing $\delta$ reduces computation cost but leads to a drop in performance. This provides flexibility to trade-off accuracy and efficiency based on application needs.

### 4.4 COMPARISON WITH TOKEN PRUNING METHODS

FLASHVLA is compared with FastV Chen et al. (2024c), a dynamic pruning baseline, and SparseVLM Zhang et al. (2024b), a structured pruning approach, on the LIBERO-Object suite under varying visual token budgets. As shown in Table 4, FLASHVLA consistently achieves comparable or better performance with lower FLOPs. At 160 tokens, for example, it attains a higher success rate (86.6% vs. 84.8%) while reducing

Table 4: Detailed Comparison between FLASHVLA, FastV Chen et al. (2024c), and SparseVLM Zhang et al. (2024b) on the LIBERO-Object task suite under various visual token budgets.

| Method / Visual token | | 256 | 192 | 160 | 128 | 96 |
|---|---|---|---|---|---|---|
| FastV | SR (%) | 86.4 | 86.2 | 84.8 | 85.2 | 85.2 |
| | FLOPs ($10^{12}$) | 1.3 | 1.0 | 0.9 | 0.7 | 0.5 |
| | Latency (ms) | 82.7 | 81.3 | 79.9 | 78.9 | 77.8 |
| SparseVLM | SR (%) | 86.4 | 85.8 | 86.0 | **85.8** | **86.0** |
| | FLOPs ($10^{12}$) | 1.3 | 1.0 | 0.9 | 0.7 | 0.5 |
| | Latency (ms) | 82.7 | 75.9 | 75.9 | 75.5 | 75.8 |
| FlashVLA | SR (%) | 86.4 | **86.6** | **86.6** | 85.2 | 83.6 |
| | FLOPs ($10^{12}$) | 1.3 | **0.7** | **0.6** | **0.4** | **0.3** |
| | Latency (ms) | 82.7 | **58.8** | **53.1** | **45.3** | **47.2** |

Table 5: Performance of FLASHVLA applied to UniVLA Bu et al. (2025), a VLA model with implicit action chunking and amortized planning, across four LIBERO tasks under different visual token budgets.

| Task / Visual token | | 256 | 192 | 160 | 128 | 96 |
|---|---|---|---|---|---|---|
| Spatial | SR (%) | 97.8 | 95.6 | 95.6 | 93.4 | 82.6 |
| | FLOPs ($10^{12}$) | 1.3 | 0.8 | 0.7 | 0.6 | 0.5 |
| Object | SR (%) | 95.8 | 95.8 | 94.8 | 86.4 | 51.2 |
| | FLOPs ($10^{12}$) | 1.3 | 0.7 | 0.5 | 0.4 | 0.3 |
| Goal | SR (%) | 95.0 | 94.8 | 95.2 | 93.2 | 94.0 |
| | FLOPs ($10^{12}$) | 1.3 | 0.6 | 0.5 | 0.4 | 0.3 |
| Long | SR (%) | 91.0 | 90.6 | 88.8 | 83.8 | 61.4 |
| | FLOPs ($10^{12}$) | 1.3 | 0.6 | 0.5 | 0.4 | 0.4 |

FLOPs from 0.85 to 0.57 ($\times 10^{12}$). While SparseVLM maintains stable SR across token budgets, its FLOPs reduction is limited compared to FLASHVLA. Overall, while both FastV and SparseVLM fall under the token pruning paradigm, a key advantage of FLASHVLA is compatibility with FlashAttention Dao et al. (2022), a memory-efficient and GPU-optimized attention kernel widely used in LLMs. This enables FLASHVLA to achieve lower memory overhead and faster inference, making it suitable for real-time deployment.

### 4.5 GENERALIZATION TO DIVERSE VLA ARCHITECTURES

To assess the generality of FLASHVLA, we apply it to UniVLA Bu et al. (2025), a VLA model with implicit action chunking and amortized planning. As shown in Table 5, FLASHVLA preserves strong performance under different token budgets, maintaining high success rates while substantially reducing FLOPs. For instance, on the Spatial task it achieves 95.6% SR with only $0.74 \times 10^{12}$ FLOPs at 160 tokens, compared to 97.8% SR with $1.31 \times 10^{12}$ FLOPs at 256 tokens. These results demonstrate that FLASHVLA generalizes effectively to architectures with latent planning structures without requiring retraining.

### 4.6 GENERALIZATION TO OTHER ENVIRONMENTS

To assess the generality of our approach, we evaluate it on VLAbench Zhang et al. (2024a), a simulated robot environment featuring more diverse and challenging tasks. Specifically, we test on the *Select Painting* task using OpenVLA with LoRA-fine-tuned weights Hu et al. (2022) provided by the authors. As shown in Table 3, although the overall success rate is low due to task difficulty, our method significantly reduces computational cost while preserving baseline performance.

## 5 CONCLUSION

We propose FLASHVLA, the first training-free and plug-and-play acceleration framework that enables action reuse in VLA models. By exploiting two forms of redundancy—temporal coherence across consecutive actions and visual token redundancy—FLASHVLA improves inference efficiency through token-aware action reuse and information-guided token pruning, reducing unnecessary computation across action steps and within inputs. Experiments on the LIBERO benchmark show that FLASHVLA reduces FLOPs by 55.7% and latency by 36.0%, with only a 0.7% drop in success rate—demonstrating practicality, effectiveness, and scalability for efficient VLA inference. Moreover, FLASHVLA generalizes to other VLA architectures such as UniVLA, maintaining high success rates with reduced FLOPs. In future work, we plan to explore additional inference acceleration techniques further tailored to the unique characteristics of VLA models.

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

# Appendix for FLASHVLA

## A    ALGORITHM FLOW OF FLASHVLA

This section provides a detailed description of the algorithmic workflow of FLASHVLA, as outlined in Algorithm 1. The overall execution is divided into two phases: initialization and iterative reasoning.

During the initialization phase (lines 1–5), the agent executes the first two steps without action reuse to establish initial context. For each of the first two frames, the model selects a subset of informative visual tokens using the strategy described in Section 3.2, performs pruned inference based on the selected tokens, executes the resulting action, and updates the *Action Memory* and *Token Memory* with the new observations and outputs.

The iterative phase begins thereafter (lines 6–19), and continues until the task is successfully completed. At each step, the **FlashTrigger** mechanism (Section 3.3) determines whether the previous action can be reused. If reuse is triggered and the last step did not already reuse an action, the model directly reuses the previous output and sets the reuse flag. Otherwise, the model selects a new visual token subset, runs pruned inference, updates both memories, and resets the reuse flag. Regardless of reuse, the action is executed in the environment and the task state is updated.

Once the task is complete, the loop exits.

---

**Algorithm 1** FLASHVLA

1:  **for** *i in range(2)* **do**
2:      ▷ Select important visual token set, according to Section. 3.2
3:      ▷ Reasoning using these important visual token sets to get action
4:      ▷ Perform the action
5:      ▷ Update *Action Memory* and *Token Memory*, according to action and visual token set
6:  **end for**
7:  **while** *Task State* **is** *False* **do**
8:      ▷ Calculate flag *Reuse Action*, according to **Flashtrigger** in Section. 3.3
9:      **if** *Reuse Action* **is** *True* and *last reuse* **is** *False* **then**
10:         ▷ Reuse last action
11:         ▷ Set flag *last reuse* as *True*
12:     **else**
13:         ▷ Select important visual token set, according to Section. 3.2
14:         ▷ Reasoning using these important visual token sets to get action
15:         ▷ Update *Action Memory* and *Token Memory*, according to action and visual token set
16:         ▷ Set flag *last reuse* as *False*
17:     **end if**
18:     ▷ Perform the action
19:     ▷ Update *Task State*
20:     **if** *Task State* **is** *True* **then**
21:         ▷ **break**
22:     **end if**
23: **end while**

---

## B   COMPUTATION COST ESTIMATION

To analyze the computational efficiency of FlashVLA, we estimate the FLOPs consumed by the Multi-Head Attention (MHA) and Feed-Forward Network (FFN) modules, which dominate the cost of Transformer-based architectures. In VLA models, visual tokens typically account for more than 80% of the input, making them the primary contributor to overall inference cost. Since the number of language prompt tokens varies across tasks, we use the FLOPs associated with visual tokens as a consistent and representative measure of complexity.

The total FLOPs are estimated as:

$$\text{FLOPs} = (1 - R) \times \left[ L_p \cdot (4nd^2 + 2n^2d + 2ndm) + (L - L_p) \cdot (4n_pd^2 + 2n_p^2d + 2n_pdm) \right] \quad (7)$$

where:

- $n$: total number of input tokens (visual + language),
- $d$: hidden dimension,
- $m$: intermediate dimension in the FFN module,
- $L$: total number of Transformer layers,
- $L_p$: layer index at which visual token pruning starts,
- $n_p$: number of visual tokens after pruning,
- $R$: action reuse rate.

By default, we set $L_p = 2$ during the prefill stage, meaning that full-token computation is retained in the first two layers to preserve early-layer representation quality. During decoding, FlashVLA reuses token selections from the prefill stage and sets $L_p = 0$. Therefore, the actual FLOPs of FlashVLA are slightly lower than the values reported in this paper, as both prefill and decoding stages are estimated using $L_p = 2$ for consistency.

## C   THEORETICAL JUSTIFICATION OF INFORMATION CONTRIBUTION SELECTION

To theoretically justify the superiority of ICS-based token selection over random sampling, we analyze the information retention in the top-$K$ selected tokens. Let $S \subset \{1, \ldots, N\}$ with $K \subset (1, N)$ be the indices of selected tokens, and let $T_S^v \in \mathbb{R}^{K \times d}$ denote the corresponding token matrix. The retained information is measured by the Frobenius norm of its projection onto the top-$r$ singular directions:

$$I(S) = \|T_S^v V_r\|_F^2 = \sum_{x \in S} \sum_{i=1}^{r} (u_{xi}\sigma_i)^2. \quad (8)$$

Maximizing $I(S)$ ensures the preservation of the dominant subspace of $\hat{T}^v$. Although $C(x)$ is defined via absolute values, it provides a greedy approximation to maximizing $I(S)$. Specifically, by the Cauchy–Schwarz inequality:

$$C(x)^2 \leq r \sum_{i=1}^{r} (u_{xi}\sigma_i)^2. \quad (9)$$

Thus, ranking tokens by $C(x)$ identifies those with high energy in the top singular directions. The retained information of the top-$K$ tokens is:

$$I(S_C) = \sum_{x \in S_C} \sum_{i=1}^{r} (u_{xi}\sigma_i)^2, \quad (10)$$

while for uniformly random selection, the expected retention is:

$$\mathbb{E}[I(S_{\text{rand}})] = \frac{K}{N} \sum_{x=1}^{N} \sum_{i=1}^{r} (u_{xi}\sigma_i)^2. \tag{11}$$

Since the top-$K$ tokens ranked by $C(x)$ dominate this global sum, we have $I(S_C) \geq \mathbb{E}[I(S_{\text{rand}})]$. Therefore, ICS-based selection guarantees better structural preservation in expectation.

## D  Experimental Setup (Details)

**Evaluation Environment.**  We evaluate FLASHVLA on the LIBERO simulation benchmark, which uses a simulated Franka Emika Panda arm and provides multimodal demonstration data, including camera observations, robot states, task labels, and delta end-effector pose actions. The benchmark contains four task suites—LIBERO-Spatial, LIBERO-Object, LIBERO-Goal, and LIBERO-Long—each with 500 expert demonstrations across 10 tasks. These suites are designed to test policy generalization under variations in spatial layouts, object types, goal specifications, and long-horizon task sequences.

**Implementation Details.**  We apply FLASHVLA to accelerate the OpenVLA model fine-tuned on LIBERO. All experiments are conducted on a single NVIDIA H100 GPU. We evaluate under different visual token configurations using three metrics: success rate (SR), inference latency, and visual-token FLOPs. Latency is measured via wall-clock time, and FLOPs are estimated following Appendix B. Real-world runtime profiling is performed with `torch.profiler`.

FLASHVLA employs a threshold-based action reuse mechanism controlled by two hyperparameters, $\varepsilon_1$ and $\varepsilon_2$. Unless specified otherwise, we set $\varepsilon_1 = 2$ and use the following $\delta$ values for different token counts: $(192, 3)$, $(160, 4.5)$, $(128, 5)$, $(96, 5.5)$. These $\delta$ values are converted to $\varepsilon_2$ using Equation 5. A full sensitivity analysis is provided in Section 4.3.

## E  FLOPs Analysis of FLASHVLA

To further analyze the efficiency of FlashVLA, we provide a detailed FLOPs breakdown across the four LIBERO task suites (Spatial, Object, Goal, and Long-horizon). This complements the main results by explicitly showing how computation savings arise from both token pruning and computation reuse.

As shown in Figure 7, FlashVLA maintains substantial efficiency improvements under different token budgets. The decomposition into effective FLOPs, reuse action, and pruned inference highlights the contribution of each component. This analysis provides further evidence that the dual-path acceleration strategy effectively reduces computational cost while preserving strong task performance.

## F  Visual Redundancy in VLA Models

To better understand the presence of visual redundancy in VLA models, we analyze the behavior of attention maps and attention scores across transformer layers. Figure 8 shows the average attention maps of VLA and VLM models across different layers. We observe that both types of models exhibit similar patterns: in early layers, attention is distributed relatively uniformly across visual tokens, while starting from the second layer, the attention maps become increasingly sparse and concentrated on fewer regions. This layer-wise transition suggests that redundancy accumulates early in the encoding process, making some tokens less informative in deeper layers.

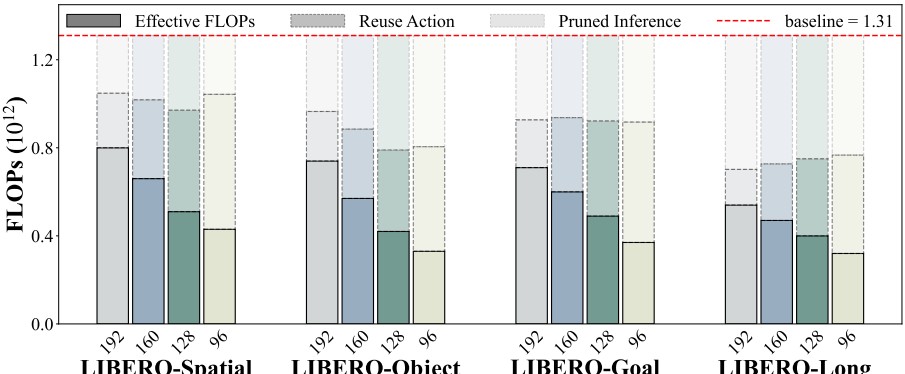

Figure 7: FLOPs breakdown of FlashVLA across four LIBERO tasks under different visual token budgets. Each bar shows the cumulative reduction in FLOPs contributed by token pruning and computation reuse. FlashVLA consistently operates below the baseline FLOPs (dashed line), demonstrating the effectiveness of the dual-path acceleration strategy.

To further quantify this sparsification effect, we examine the attention scores of visual tokens computed from the last transformer layer. Specifically, we extract the raw attention weights from the model outputs as follows:

```
layer_attention = layer_outputs[1]
layer_attention_avg = torch.mean(layer_attention, dim=1)[0]
attention_score = layer_attention_avg[-1]
```

Here, the attention weights are averaged over all heads, and the final row corresponds to the attention received by each visual token when queried by the final position (e.g., the action token or decoder query). We use this vector as the attention score distribution.

As shown in Figure 9, the attention scores are nearly uniform across token positions in the first two layers. However, starting from the second layer, we observe a clear increase in variance, with attention values increasingly concentrated on a small subset of tokens. This indicates a growing redundancy among visual tokens in deeper layers—a phenomenon also observed in recent studies on VLMs Chen et al. (2024c). These observations provide empirical evidence for the existence of token-level redundancy in VLA models and motivate our token pruning strategy.

# G  ADDITIONAL EXPERIMENTAL DETAILS

## G.1  LIBERO SIMULATED ENVIRONMENT BENCHMARK

LIBERO is a novel benchmark designed for studying knowledge transfer in multitask and lifelong robot learning. It addresses the challenge of benchmarking knowledge transfer capabilities in robot learning systems, with a focus on manipulation tasks that require both declarative knowledge (about objects and spatial relationships) and procedural knowledge (about motion and behaviors) .

LIBERO provides four main task suites, each designed to evaluate different aspects of knowledge transfer:

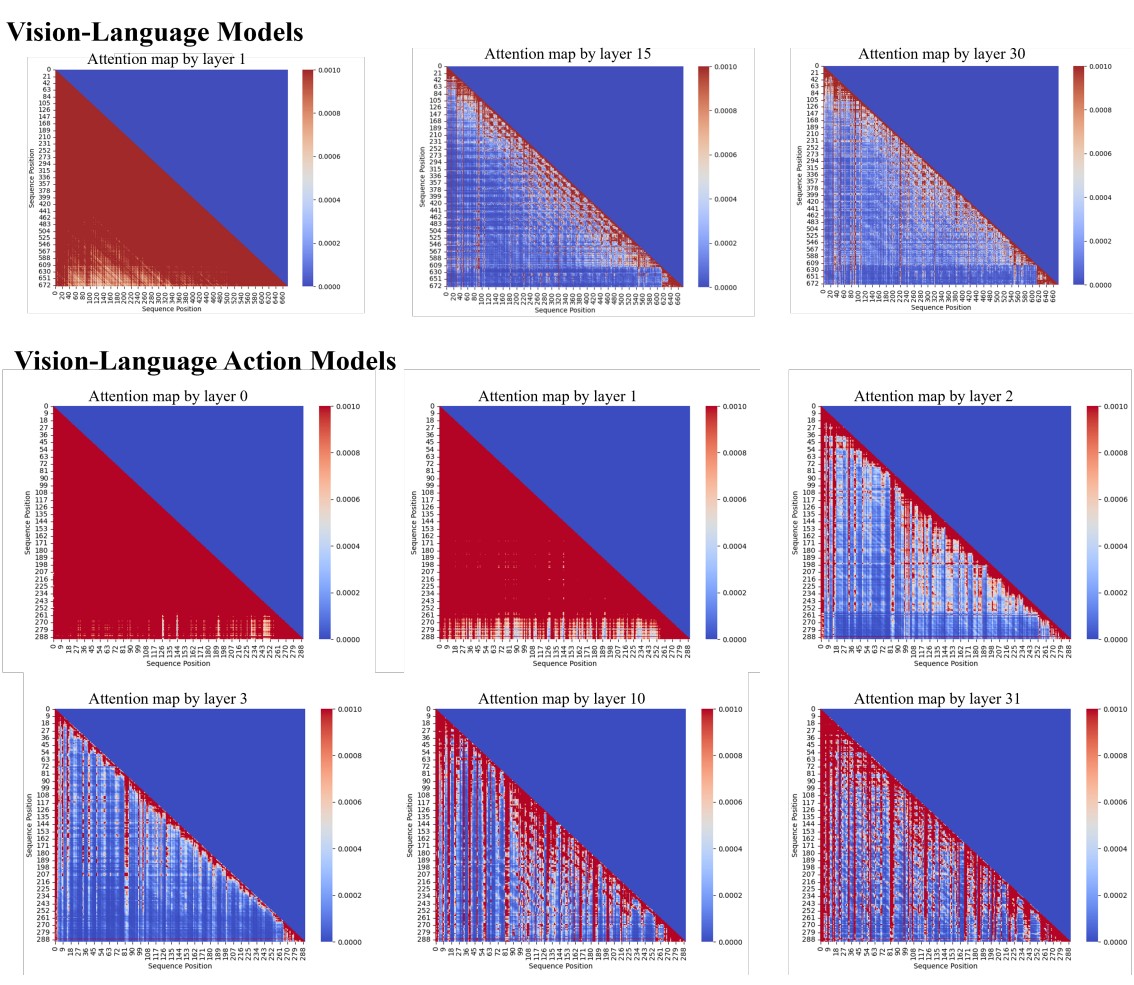

Figure 8: **Attention Map**: Layer-wise attention map visualizations in VLA and VLM models. Both models exhibit uniform attention distribution in the first layer, while attention becomes increasingly sparse from the second layer onward. This pattern suggests growing redundancy in token interactions, motivating token pruning strategies in deeper layers

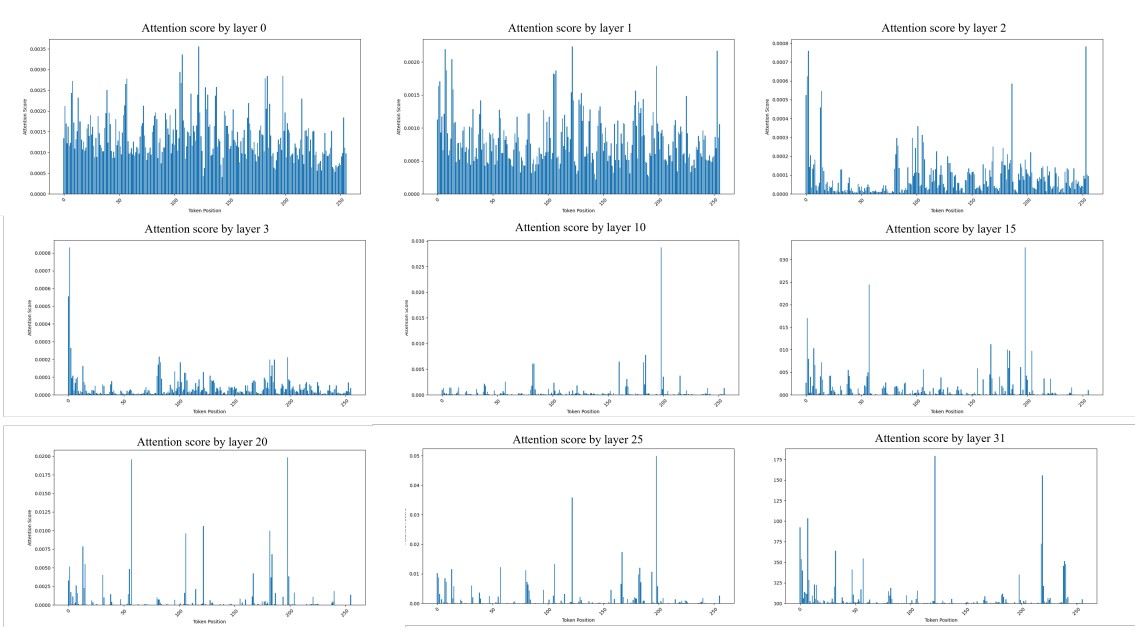

Figure 9: **Attention Score**: Attention score distributions across transformer layers in a VLA model. The scores are computed by averaging attention weights over heads and selecting the attention received by each token from the final query position. The results show increasing sparsity from the second layer onward, where attention becomes concentrated on a small subset of tokens.

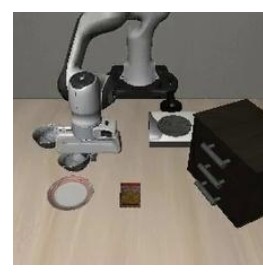 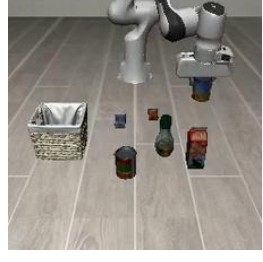 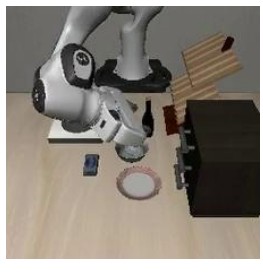 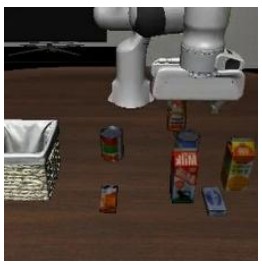

**LIBERO-Spatial**    **LIBERO-Object**    **LIBERO-Goal**    **LIBERO-Long**

Figure 10: Sample Frame of Four Main Task Suites.

**LIBERO-Spatial**  It contains 10 tasks that focus on transferring knowledge of spatial relationships. These tasks require robots to understand the spatial relationship between different objects and use this knowledge to complete the task. For example, the robot need to place objects according to a certain spatial layout, or navigate to the target position according to spatial clues in a complex environment. Through these tasks, its ability to master and apply the knowledge of spatial relationship is investigated.

**LIBERO-Object**  It consists of 10 tasks that require transferring object-related knowledge. Robots are expected to identify different objects, comprehend their attributes (e.g., color, shape, material) and functions (e.g., tool usage, container functionality), and manipulate the objects accordingly. Examples include classifying objects based on their attributes or utilizing tools to perform specific tasks. These tasks serve to measure the robot's capability to transfer knowledge related to objects.

**LIBERO-Goal**  It has 10 tasks that emphasize transferring goal-oriented knowledge. Robots must precisely comprehend the task objectives, determine the essential steps and strategies for achieving them. For example, it should be able to accurately prioritize goals in multi - task scenarios or break down complex goals into manageable sub - goals and accomplish them step by step. The evaluation aims to assess the robot's ability to transfer and apply goal - oriented knowledge effectively.

**LIBERO-Long**  It has 10 tasks primarily designed to evaluate the robot's knowledge transfer ability over extended learning periods. These tasks typically involve learning and integrating knowledge across multiple tasks. The investigation focuses on whether the robot can effectively apply the experience, skills, and knowledge acquired from previous tasks to new subsequent tasks, thereby achieving continuous performance enhancement and improved adaptability. Furthermore, it emphasizes the accumulation, updating, transfer, and application of knowledge throughout the long - term learning process.

## H    LIMITATIONS AND FUTURE WORKS

We propose FLASHVLA, the first training-free and plug-and-play acceleration framework that enables action reuse in VLA models. Although our approach maintains the performance of the model while greatly reducing the amount of modeling operations and the actuallatency, there are limitations and shortcomings in our approach. First of all we have only tested in a simulated environment, lacking further validation in the real world. Second, we only performe experimental validation of a single-arm robot. In the future, we will further validate the advantages of our approach in extending our method to more robotic arms with more degrees of freedom.

# I    USE OF LARGE LANGUAGE MODELS (LLMS)

In accordance with the ICLR policy on LLM usage, we disclose that we used an LLM (ChatGPT by OpenAI) only after completing the full manuscript draft, and solely for surface-level proofreading: correcting grammar, punctuation, and minor phrasing for clarity and consistency. The LLM did not contribute to research ideation, problem formulation, method or experiment design, data collection or labeling, or analysis. Every suggested edit was manually reviewed and selectively adopted by the authors.

We understand and accept full responsibility for all content written under our names, including any text that may have been revised with LLM assistance. We took care to avoid plagiarism and factual errors, and we did not provide the LLM with proprietary or personally identifiable data beyond de-identified manuscript excerpts necessary for proofreading. The LLM is not an author or contributor under ICLR authorship criteria.

