# OpenReview forum: "Think Twice, Act Once: Token-Aware Compression and Action Reuse for Efficient Inference in Vision-Language-Action Models"
_ICLR.cc/2026/Conference — ICLR 2026 Conference Withdrawn Submission_

### Official Review · Reviewer_kQ7p · 2025-10-26

**Soundness:** 2
**Presentation:** 3
**Contribution:** 2
**Rating:** 4
**Confidence:** 4

**Summary:**

This paper presents FLASHVLA, a training-free and plug-and-play acceleration framework for Vision-Language-Action (VLA) models. Instead of architectural re-design or retraining, the method targets two types of redundancy: (1) temporal similarity between consecutive actions and (2) redundancy among visual tokens. To address these, the authors propose a token-aware action reuse mechanism and an information-guided token pruning strategy based on singular value decomposition (SVD) and information contribution scores (ICS).
Experiments on LIBERO, UniVLA, and VLAbench show that FLASHVLA reduces FLOPs by 55.7% and latency by 36.0%, with only 0.7% performance drop. While the idea is appealing and practical, the methodology mainly combines known techniques (e.g., token pruning, reuse heuristics) with moderate novelty.

**Strengths:**

- Proposes a simple yet effective framework to reduce redundant computations in VLA inference.

- Training-free and compatible with FlashAttention, enabling easy integration into existing models.

- Demonstrates strong empirical results on multiple benchmarks.

- Includes detailed ablation and sensitivity analyses.

**Weaknesses:**

- The approach primarily combines known concepts (token pruning, reuse heuristics) without strong theoretical advancement.

- The method is evaluated mostly in simulated environments; real-robot deployment results are missing.

- Performance may depend on manually tuned thresholds; no adaptive mechanism is proposed.

- Works such as VLA-Cache [1], TinyVLA [2], EfficientVLA [3] should be discussed and experimentally compared.
- Some related token pruning methods should also be discussed and compared, such as AIM
[4], DART [5], VisPruner [6]


[1] Xu, Siyu, et al. "Vla-cache: Towards efficient vision-language-action model via adaptive token caching in robotic manipulation." arXiv preprint arXiv:2502.02175 (2025). \
[2] Wen, Junjie, et al. "Tinyvla: Towards fast, data-efficient vision-language-action models for robotic manipulation." IEEE Robotics and Automation Letters (2025). \
[3] Yang, Yantai, et al. "EfficientVLA: Training-Free Acceleration and Compression for Vision-Language-Action Models." arXiv preprint arXiv:2506.10100 (2025). \
[4] Zhong, Yiwu, et al. "Aim: Adaptive inference of multi-modal llms via token merging and pruning." Proceedings of the IEEE/CVF International Conference on Computer Vision. 2025. \
[5] Wen, Zichen, et al. "Stop looking for important tokens in multimodal language models: Duplication matters more." arXiv preprint arXiv:2502.11494 (2025). \
[6] Zhang, Qizhe, et al. "Beyond text-visual attention: Exploiting visual cues for effective token pruning in vlms." Proceedings of the IEEE/CVF International Conference on Computer Vision. 2025.

**Questions:**

- How stable is the reuse mechanism under long-horizon or dynamic-scene tasks?
- Can the reuse threshold be made adaptive, e.g., based on uncertainty or temporal confidence?
- Could the authors report real-world timing (wall-clock) results beyond FLOPs estimation?

---

### Official Review · Reviewer_qWBe · 2025-10-30

**Soundness:** 3
**Presentation:** 2
**Contribution:** 2
**Rating:** 4
**Confidence:** 4

**Summary:**

This paper proposes a training-free and plug-and-play acceleration framework named FlashVLA. The proposed method features a token-aware action reuse mechanism and a visual token selection strategy that integrate seamlessly with Flash Attention to avoid redundant computation. The experiments on the LIBERO simulation benchmark show that inference latency and the FLOPs of visual token computation are both decreased considerably without significantly sacrificing the success rate.

**Strengths:**

1. Substantial reduction in FLOPs and inference latency without additional fine-tuning;
2. The method is straightforward and can be integrated with models that use Flash Attention for inference.

**Weaknesses:**

All the experiments are conducted on simulation manipulation benchmarks (LIBERO, VLABench). It lacks validation on tasks that involve highly dynamic actions (requiring frequent and rapid changes in actuators) and rapidly changing visual scenes (with significant perturbations in objects and background).

**Questions:**

As we know, real-world tasks are prone to visual perturbations and sensor noise, while robotic arms face a sim-to-real gap. Could the action reuse mechanism significantly compromise the execution accuracy of the robotic arm? How effective can visual token reduction remain in the face of real-world visual variations?

---

### Official Review · Reviewer_GNjz · 2025-10-31

**Soundness:** 3
**Presentation:** 2
**Contribution:** 2
**Rating:** 4
**Confidence:** 3

**Summary:**

This paper proposes FLASHVLA, a training-free acceleration framework for Vision-Language-Action (VLA) models. The method is motivated by the identification of two key redundancies: high similarity across consecutive action steps and high redundancy in visual tokens. Experiments on the LIBERO benchmark show that FLASHVLA can reduce FLOPs and latency.

**Strengths:**

- FLASHVLA can be directly plugged into existing VLA models (e.g., OpenVLA, UniVLA) without retraining, which makes it both practical and reproducible.

- The extensive ablation studies on # of Tokens, different modules and diverse VLA architectures make the results convincing.

**Weaknesses:**

- Both the similarity across consecutive action steps and across visual tokens have been well-studied. “The first training-free and plug-and-play acceleration framework that enables action reuse in VLA models” also needs more justification.

- The method's hyperparameters appear to require careful, per-setting tuning, which may limit reproducibility. Specifically, the parameter $\delta$, which controls the token set stability threshold $\epsilon_2$, is set to different values for each token budget (e.g., 3, 4.5, 5, 5.5).

- The experiments should be further strengthened. First, the experiments on other VLA acceleration baselines and other benchmarks should be included. Second, the evaluation is confined to simulation. While the simulation results on LIBERO are strong, the paper lacks validation on a real-world robot.

**Questions:**

- How sensitive is performance to the thresholds ε1 (action angle) and ε2 (token overlap)?  Could these parameters be adaptively learned or adjusted at runtime?

- In scenarios with discontinuous or abrupt motion (e.g., contact manipulation), does the reuse mechanism still function reliably, or could it mis-predict stable states?

- There seems to be a critical contradiction in the description of the "FlashTrigger" mechanism (Section 3.3). Could the authors please clarify the action reuse trigger logic in Equation 6? Is the condition $\alpha(s) > \epsilon_1$ a typo, and should it be $\alpha(s) < \epsilon_1$? The current formulation seems to contradict the stated goal of reusing actions in "stable areas" where the angle change is small.

- The visual token selection strategy is applied after the first two layers ($L_p=2$) in the prefill stage. What is the rationale for this specific depth? How sensitive is the model's performance to this $L_p$ parameter?

---

### Note · Authors · 2025-11-12

I have read and agree with the venue's withdrawal policy on behalf of myself and my co-authors.